# Nano-Dry-Melting: A Novel Technology for Manufacturing of Pharmaceutical Amorphous Solid Dispersions

**DOI:** 10.3390/pharmaceutics14102145

**Published:** 2022-10-09

**Authors:** Malin Hermeling, Christoph Nueboldt, Roman Heumann, Werner Hoheisel, Joerg Breitkreutz

**Affiliations:** 1INVITE GmbH, 51368 Leverkusen, Germany; 2Institute of Pharmaceutics and Biopharmaceutics, Heinrich Heine University Duesseldorf, 40225 Duesseldorf, Germany; 3Bayer AG, 51368 Leverkusen, Germany

**Keywords:** amorphous solid dispersion, amorphization, wet stirred media milling, spray drying, nanoparticles, nanocomposition, niclosamide

## Abstract

Amorphous solid dispersions (ASD) are one of the most prominent formulation approaches to overcome bioavailability issues that are often presented by new poorly soluble drug candidates. State-of-the art manufacturing techniques include hot melt extrusion and solvent-based methods like spray drying. The high thermal and mechanical shear stress during hot melt extrusion, or the use of an organic solvent during solvent-based methods, are examples of clear drawbacks for those methods, limiting their applicability for certain systems. In this work a novel process technology is introduced, called Nano-Dry-Melting (NDM), which can provide an alternative option for ASD manufacturing. NDM consists of a comminution step in which the drug is ground to nanosize and a drying step provides a complete amorphization of the system at temperatures below the melting point. Two drug–polymer systems were prepared using NDM with a wet media mill and a spray dryer and analyzed regarding their degree of crystallinity using XRD analysis. Feasibility studies were performed with indomethacin and PVP. Furthermore, a “proof-of-concept” study was conducted with niclosamide. The experiments successfully led to amorphous samples at temperatures of about 50 K below the melting point within seconds of heat exposition. With this novel, solvent-free and therefore “green” production technology it is feasible to manufacture ASDs even with those drug candidates that cannot be processed by conventional process technologies.

## 1. Introduction

The current trend of drug development is towards drug molecules with high molecular weight and poor hydrophilicity [1]. As an undesired side effect, the solubility in gastrointestinal fluids is negatively impaired. Thus, new drug molecules often show high efficacy but insufficient bioavailability due to limited solubility. Recent efforts to overcome such limitations include alteration of the chemical nature (like salt formation, cocrystals) or modification of the physical solid state. A very prominent approach for the latter one is called “Amorphous Solid Dispersion” (ASD), in which the drug is molecularly dispersed in a water-soluble carrier matrix typically formed by a polymer. In an ASD the high energy amorphous state of the drug is stabilized by the surrounding matrix, which may facilitate the dissolution process in the aqueous environment of the gastro-intestinal tract [2].

The state-of-the-art manufacturing routes can be roughly divided into two categories: (a) fusion-, and (b) solvent-based methods. Both routes start with the disruption of the crystal lattice. The drug molecules are then dispersed and entrapped within the polymer carrier. This is done by either fast cooling or by removal of solvent(s) [3].

In fusion-based methods, the components are mixed and heated up to (or close to) the melting point of the components. The low viscosity at this state enables the distribution of drug molecules throughout the carrier. Fusion-based methods typically differentiate by the process of heating (e.g., using thermal heat, microwaves or ultrasonic waves) or the mechanical input [4]. The most prominent example for fusion-based methods is hot melt extrusion (HME), where the product is subjected to high thermal and mechanical stress. However, this exposure can lead to degradation of the drug molecule and also of the carriers involved [5]. In solvent-based methods, all the components are dissolved in an organic solvent. This is removed in the subsequent rapid drying process, which enables the entrapment of the drug molecules by the carrier. In solvent-based methods, moderate temperatures and lower mechanical load are applied during the process. However, there is the major disadvantage concerning the need for usually huge amounts of organic solvents. It can be difficult to identify a common solvent for the mostly hydrophilic carrier and the hydrophobic drug. This solvent must provide sufficient solubility for both components which is necessary to realize economic production. Later, all residual solvents must be removed, which often leads to the need for a secondary drying step. In addition, health hazards for the operator team must be considered when large scale organic solvents are used as well as other safety issues like explosion risk. Moreover, the use of organic solvents is not only costly because of the required collection and recycling but is also opposing the worldwide trend towards sustainability in production (“green chemistry” [6]). Solvent-based methods include spray drying (SD), fluidized bed drying or rotary drying. The methods can be distinguished by their droplet formation (e.g., via electric charge in electrospraying vs. via pressure in conventional SD or the rate of the removal of the solvent (rotary drying vs. SD) [5,7]. There are further methods that induce amorphization by mechanical impact like cryomilling or kneading, but these methods do not consistently fit into the two categories discussed. The challenge for these methods is to reach a complete homogenization to avoid the presence of nuclei for later phase separation and recrystallization [8]. Melt evaporation provides another exception, where the approach was meant for drugs with very high melting points. Here the drug is dissolved in a suitable solvent and then incorporated into the carrier matrix in its low-viscous state (as melt or above the glass transition temperature) [5].

Each method presents advantages and disadvantages as shown in Table 1. Hence, not all manufacturing techniques are suitable for all substance systems. Therefore, specific requirements regarding the physicochemical properties of the drug molecules can be formulated for each manufacturing technique [3]. Table 1 lists general advantages, disadvantages and requirements for the two groups that are located in the rows “solvent-based” methods and “fusion-based” methods. Items presented in the rows below should be seen as additional aspects for the specific technique.

During process development, the chemical nature of the drug molecule can provide challenges for choosing suitable materials and methods. There are molecules which are sensitive to heat or mechanical stress. Other drug candidates exhibit high melting points at which conventional polymers are already degrading, making fusion-based methods unsuitable. Some drug candidates have very low solubility in common organic solvents, so that alternative organic solvents would be required that are more expensive or less suitable for pharmaceutical use due to their toxicity. Furthermore, the hydrophobic nature of the drug and the mostly hydrophilic nature of the commonly used polymers require opposed solvent properties. Sihorkar et al. described the missing spot while choosing an implemented industrial process route for the ASD formulation of new drug candidates: What to do with drugs that have high melting points and low organic solubility [9]?

**Table 1 pharmaceutics-14-02145-t001:** Process technologies for solvent- and fusion-based ASD manufacturing methods: advantages and disadvantages.

Process Technology	Advantages	Disadvantages	Requirements for Processability
Solvent Based Method			
Solvent-based in general	Moderate temperature, Prevention of degradation [7], Molecular level mixing [10],	Organic solvent: HSEQ * risk, [7] Economic burden, secondary drying step required [7], Expensive [10], Phase separation possible [10]	Solvent with sufficient drug solubility, low toxicity, high volatility and low explosion risk [7]
Spray Drying (SD)	Easy scale up [7], Single step, Control of powder shape/form, Continuous batch manufacturing [10]	-	-
Fluidized bed granulation/ layering	Products exhibit good flowability and tableting properties [11], Reduction in additional downstream processing steps (Coating, Granulation), Can prevent stability problems [3]	Limited drug load [11]	Core required
Co-precipitation	Easy scale up [5], Higher PSD* than SD particles, Superior compaction profile, High drug load possible [3]	Post-processing required (filtering, drying) [5], Risk of recrystallization during supersaturated state in antisolvent [3]	Antisolvent for all components required [12], Antisolvent and solvent must be miscible [3]
Freeze Drying Spray-Freeze Drying	High robustness and reliability, Minimal thermal stress, Minimal risk of phase separation [7], Highly porous particles enabling fast dissolution [3]	Expensive, Large equipment, Cryoprotective may be required [7]	Sufficient drug solubility in water or inorganic solvent miscible with water [7]
Processes that include supercritical fluids (SCF)	Can generate very small particles, Low temperatures, Not dependent on organic solvent, Health and environmentally friendly [10], Low production costs [3]	Difficult to scale up, Cost-intensive equipment [7], Low solubility of most pharmaceutic compounds in CO_2_ [10]	-
**Fusion-Based Method**			
Fusion-based in general	No solvent required, No drying required [7]	High thermal load, Possible degradation [7]	Drug polymer miscibility, Thermostable substances [7]
Hot Melt Extrusion (HME)	High throughput, Low costs [13], Continuous manufacturing, Furrow mixing of components, Easy scale up, Drug release can be tailored [7], Shaping of product form [10]	High shear stress, Varying residence time, Miscibility determines process temperature [13]	Polymer of processible viscosity (low T_g_ *) Often: plasticizer(s) required
Melt Agglomeration	Use of standard granulation equipment, High variability in batch size, Useful for water sensitive drugs [7]	-	Carrier required [7]
Milling	Less thermal load (than HME), Included PSD * reduction	High mechanical stress, Degree and robustness of amorphization low/limited [7], Drying necessary [5], Typically only employed in lab scale [3]	-
Kinetisol^®^	Highly reduced processing times compared to HME (20s), Lower temperatures (than HME) [3,11], High drug load possible [3]	High mechanical load, Less stable product, High risk of drug degradation [3]	-

* HSEQ = Health, Safety, Environment and Quality, PSD = particle size distribution, T_g_ = glass transition temperature.

### Theoretical Background of Nano-Dry-Melting

The new proposed process concept called “Nano-Dry-Melting” (NDM) could offer an alternative route for the challenging drug candidates as well as an improved option for already established ASD products [14]. As the term already suggests, NDM includes the formation of drug nanoparticles which are the starting material for the process. The required nanoparticles can be produced by a precipitation (bottom-up) or by a comminution (top-down) process depending on the materials involved. Comminution of drug crystals is usually accomplished either by high pressure homogenization or by wet media milling [15]. Both processes result in aqueous suspensions of drug nanoparticles that are ideally stabilized by the same polymer required for the subsequent ASD. This means the comminution process can be performed with the drug–polymer ratio that is used for the ASD, or at a higher drug–polymer ratio while additional polymer may be added afterwards. After the comminution step the aqueous suspension is dried at elevated temperature, which is typically a few 10 K above the glass transition temperature of the polymer (T_g_). During this second “Dry-Melting” step (DM), first water is evaporated and after drying the stabilizing polymer acts as the carrier matrix that the drug particles most probably will dissolve into, forming a homogenous amorphous solid solution [14]. This dissolution of the drug into the polymer (DisP) is possible even below the melting point of the drug substance due to the exergonic impact of the mixing between polymer and drug, which leads to melting point depression [16].

Particle size reduction is known to have a positive effect on the dissolution kinetics. The mass rate of dissolution, $\frac{\mathrm{dM}}{\mathrm{dt}}$ can be described by the Noyes–Whitney Equation (Equation (1), where D, A, h, ${c\;\mathrm{and}\;c}_{s}$, represent the diffusion coefficient, the total surface area, the diffusion layer thickness, the current concentration and the saturated concentration [17]:(1)$$\frac{\mathrm{dM}}{\mathrm{dt}}=\frac{\mathrm{DA}\left(c_{s}-c\right)}{h}$$

The size reduction results in the increase in the surface area which is inversely proportional to the particle diameter (d) for the same total mass. Furthermore, in case of very small particles below 50 µm, the thickness of the diffusion layer is linear proportional to the particle size, while for larger particles, it was found to be constant at about 30 µm [18,19]. This means that by decreasing d below 50 µm, h will decrease as well and accelerate $\frac{\mathrm{dM}}{\mathrm{dt}}$ via two factors, A and h. Galli analyzed the dissolution behavior of drugs in liquid media at sizes between 0.5 and 6 µm and found a quadratic proportionality between dissolution rate and particle diameter [20]. Galli’s data show that there are hints that the diffusion layer thickness decreases even faster than linear for particles smaller than about 700 nm. Thus, suggesting a proportionality of $\frac{\mathrm{dM}}{\mathrm{dt}}$ to d that is stronger than inverse quadratic. Specifically for ASDs, the change in the dissolution rate is examined in a recent work by Seiler et al. [13]. The authors indicated that the time for DisP has a quadratic trend relative to the particle size at sizes ranging between 80 µm and 300 µm. DisP was completed within an hour at sizes that are still a thousand times larger than the ones of nanoparticles. The results of Seiler et al. suggested an already quadratic increase in the dissolution rate at sizes above 50 µm in cases of the dissolution in a polymer. Galli, on the other hand, analyzed the dissolution of drugs in aqueous solutions. Both works suggest an immense increase in the dissolution rate when particles are scaled down to nanosize in magnitudes of at least quadratic impact.

Furthermore, the size reduction will also lead to a decrease in the necessary diffusion path lengths that the dissolved molecules must travel: the larger the number of nanoparticles $\left(n_{\mathrm{nanoparticle}\;}\right)$ that is required to maintain the same drug load $\left({\mathrm{DL}}_{\mathrm{vol}.-\%}\right)$ while reducing the size $\left(d_{\mathrm{nanoparticle}}\right)$ has to distribute evenly throughout the same volume of polymer. Thus, creating a composition with lower distances between the individual particles (Equations (2) and (3)):(2)$${\mathrm{DL}}_{\mathrm{vol}.-\%}=n_{\mathrm{nanoparticle}}V_{\mathrm{nanoparticle}}=n_{\mathrm{microparticle}}V_{\mathrm{microparticle}}$$
(3)$$\frac{n_{\mathrm{nanoparticle}}}{n_{\mathrm{microparticle}}}={\left(\frac{d_{\mathrm{micro}}}{d_{\mathrm{nano}}}\right)}^{3}$$

The mean distance between drug particles throughout a composition can be calculated using a cubic grid as a simplified model in which the drug particles are positioned at the corners (Figure 1). Even though this is a simplified model it enables the visualization of the basic principles that also holds true for randomly but homogeneously distributed particles in “real” composites. There are three possible distances between the centers of those particles: the distance to the direct neighbor (x); the distance to the planar diagonal neighbor (x√2); and the distance to the cubic diagonal neighbor (x√3).

Per cubic grid there is a single spheric drug particle (eight corners with each 1/8 of a sphere). The volume of the sphere can be correlated to the volume of the cube with the volumetric drug load, ${\mathrm{DL}}_{\mathrm{vol}.-\%}$ (Equation (4)). By converting, the direct distances, x, can be connected to d of one sphere, d (Equation (5)). With this equation, the distance between the centers of different spheres is calculated in dependance of d:(4)$${\mathrm{DL}}_{\mathrm{vol}.-\%}x³=\frac{4}{3}{\left(\frac{d}{2}\right)}^{3}\pi$$
(5)$$\Leftrightarrow x=d\sqrt[{3}]{\left(\frac{\pi }{6{\mathrm{DL}}_{\mathrm{vol}.-\%}}\right)}$$

Since the first molecules leaving the crystal lattice are located at the surface of the spheres, the distance to travel is not from “center to center” but “surface to surface”. The distance between the centers can therefore only be assumed as the absolute maximum of distances to travel. Only the innermost molecules, leaving the crystal lattice at last, would have to travel from center to center. However, this region would be already filled with other already dissolved and diffused molecules. The length between the centers $L_{\mathrm{mean}}$ can be calculated according to Equation (6) and could be assumed to be the worst-case diffusion pathway. More realistically, the distance between the surfaces, the “interparticular spacing” $({\mathrm{IPS}}_{\mathrm{mean}}$) can be calculated directly using Equation (7) [21]. Both equations present the mean values of the direct distance, the distance diagonal on a planar level and on a cubic level:(6)$$L_{\mathrm{mean}}=d\sqrt[{3}]{\left(\frac{\pi }{6{\mathrm{DL}}_{\mathrm{vol}.-\%}}\right)}$$
(7)$${\mathrm{IPS}}_{\mathrm{mean}}=d\left[\sqrt[{3}]{\left(\frac{\pi }{6{\mathrm{DL}}_{\mathrm{vol}.-\%}}\right)}-1\right]$$

The diffusion time required to conquer these distances can be roughly estimated using the mean square displacement according to Equation (8). L corresponds to the diffusion path length, n presents the number of dimension (3 in case of a cubic system), D represents the Diffusion coefficient and t the required time. As stated in Equation (9), the time is proportional to the quadratic diffusion path length:(8)$$L^{2}=2\mathrm{nDt}$$
(9)$$\Leftrightarrow t=\frac{L²}{2\mathrm{nD}}\;$$

The decreased distances in case of a nanocomposition therefore result in much shorter time periods for the uniform distribution of the drug molecules. The driving force in such a homogenization process is, beside the random walk of molecules, probably the concentration gradient. This is not considered in the mean square displacement. However, since the final concentration of drug molecules within each composition should be the same, this effect should be equally strong in case of a micro- or a nanocomposition. The reduction in the particle size into nanoscale therefore has two positive impacts on DisP: the increase in the surface area with a simultaneous decrease in the diffusion layer accelerates the dissolution rate of the drug particles, and in addition, a reduction in the time required for the homogeneous distribution of the drug molecules, which arises from the decreased interparticular spacing. Both result in a considerably faster formation of an ASD in which the drug is uniformly molecularly distributed.

## 2. Materials and Methods

### 2.1. Materials

For preliminary experiments indomethacin, precisely the gamma-polymorph, (CAS 53-86-1) by abcr GmbH with a purity of 99% was used. Indomethacin exhibits a melting temperature of 161 °C and a glass transition temperature of 45 °C (both determined via DSC). For all experiments niclosamide anhydrate (CAS 50-65-7) by Acros Organics with 97.5+% purity was used. Its aqueous solubility is 13–15 mg/L at 25 °C, making it a typical BCS class II agent. Furthermore, its melting point is 230 °C. In Differential scanning calorimetry (DSC) experiments, a very unstable pure amorphous film was formed, which quickly recrystallized even when cooled with liquid nitrogen, making it a ”GFA class I” candidate according to Baird et al. (data not shown) [22] Furthermore, niclosamide monohydrate (CAS 73360-56-2) was synthesized after Van Tonder et al. [23]. The analytical standard of niclosamide monohydrate was also purchased by Acros organics for reference measurements. Polyvinylpyrrolidone, PVP K12 and K25 (CAS 9003-39-8) by BASF were used as pharmaceutically relevant carrier polymer for the ASD production. PVP K12 (T_g_~110 °C) was used for the Indomethacin ASD and PVP K25 for the niclosamide experiments. PVP K25 is known to have stabilizing properties during nanomilling procedures but shows a high glass transition temperature (T_g_~160 °C). Sodium dodecyl sulfate (SDS) from abcr GmbH (CAS 151-21-3) with 99 wt.-% purity was used as surface-active agent. The chemical structures are shown in Figure 2.

### 2.2. Methods

#### 2.2.1. Technical Implementation of NDM Process

The NDM process was implemented by applying two unit operations. First, Nanomilling of an aqueous drug slurry in a wet stirred media mill (WSMM) was performed. A subsequent DM step, which includes the solvent evaporation and the DisP, was performed using a standard spray dryer (Figure 3).

#### 2.2.2. Milling

A microsuspension was prepared by dissolving polymer (and surfactant) in water. The solution was stirred for 3 h until the polymer was completely dissolved. Then, the drug was added and the suspension was stirred for at least 3 h, but a maximum of overnight, until the suspension appeared homogeneous. The used magnetic stirrer was the RCT basic by IKA; Staufen Germany. The resulting compositions are shown in Table 2.

For the indomethacin suspension, the milling took place in the planetary ball mill “Pulverisette 5” by Fritsch, Idar-Oberstein Germany. The small sample unit (23 mL) was used and 60 vol.-% of the bulk volume was filled with beads made of yttrium-stabilized zirconium oxide (SiLibeads^®^ type ZY-E, 0.4–0.6 mm diameter by Sigmund Lindner, Warmensteinach, Germany). The suspension was milled for 90 min at 400 rpm. The mill was not temperature controlled. For the niclosamide suspensions the milling took place in a WSMM: Picoline, with Picoliq unit by Hosokawa Alpine, Augsburg Germany, used in batch mode. The milling chamber with a listed volume of 90 mL was used. The actual volume was reviewed by measuring the amount of water that fits in the chamber. The resulting volume was 100 mL, with 80 vol.-% of the bulk volume filled with beads made of yttrium-stabilized zirconium oxide (SiLibeads^®^ type ZY-E, 0.4–0.6 mm diameter by Sigmund Lindner, Warmensteinach, Germany). The temperature of the milling chamber was controlled by an external cooling system and kept below 45 °C. Rotational speed was set to 4000 rpm and all samples were milled for 45 min.

#### 2.2.3. Spray Drying

For SD a Lab Spray Dryer B-290 by Büchi, Essen Germany, was used. Air was used as drying gas and nitrogen gas for the atomization. Samples were collected using the small cyclone. The feed rate was 1.5 g/min. In case of the indomethacin suspension, the aspirator was set to 100%, which equals a drying gas flow of 35 m³/h. The rotameter for nitrogen flow was set to 40 mm which is equivalent to a gas flow of 0.473 m³/h and a pressure drop of approx. 0.41 bar. For the niclosamide suspensions, the aspirator was set to 60%, which equals a drying air gas flow of 25 m³/h. The rotameter for nitrogen flow was set to 50 mm which is equivalent to a gas flow of 0.601 m³/h and a pressure drop of approx. 0.75 bar.

#### 2.2.4. Preparation of Physical Mixture

For obtaining physical mixtures (PM), the same composition of components was placed inside a 20 mL vial and mixed using a 3D shaker mixer, “Turbula” by Willy A. Bachofen AG, Muttenz Switzerland, for 45 min at 80 rpm.

#### 2.2.5. Experiments with the Kofler-Bench

An aliquot of the analyzed suspension was placed onto a Kofler-bench which exhibits a temperature gradient from right to left: 50 °C on the right up to 200 °C on the left. The suspension was left to dry completely before visually inspected.

#### 2.2.6. Freeze-Drying

The suspension was dried using a freeze dryer Epsilon 2-6D LSCplus by Martin Christ Gefriertrocknungsanlagen GmbH, Osterode am Harz Germany. The sample stage was precooled to −40 °C before the sample was placed inside. Freezing took place at −40 °C and 1000 mbar for 180 min. Main drying took place at −40 °C and 0.2 mbar for 130 min, followed by drying at 0 °C and 0.2 mbar for 720 min. Secondary drying was performed at 20 °C and 0.2 mbar for 600 min, followed by a drying step for 610 min at 30 °C and 0.05 bar. The capability to dry nanosuspensions by this procedure was already proven in previous work. Vials (10 mL) specially made for freeze-drying were filled with suspension to a height of <1 cm (~0.6 g Suspension).

#### 2.2.7. Determination of Process Conditions

The investigated process characteristics are the product temperature as well as the heat exposure time. In general, the product residence time in a spray dryer can be estimated using an already established assumption that the time is as long as the residence time of the drying gas [24]. Equation (10)) was used for the calculation of the residence time $t_{r}$, in which $V_{\mathrm{DC}}$ corresponds to the volume of the drying chamber and ${\dot{V}}_{\mathrm{air},\mathrm{in}}$ to the volumetric drying gas flow. The used height of the machine is 47 cm, the diameter 15.8 cm [25]:(10)$$t_{r}=\frac{V_{\mathrm{DC}}}{{\dot{V}}_{\mathrm{air},\mathrm{in}}}$$

In order to calculate the temperatures in the DM-process inside the spray dryer, the following assumptions were made:
The heat transferred to the suspension will be consumed for the evaporation of water first, which is an endothermic process. It can therefore be assumed that the solid product will only heat up beyond 100 °C after most of the water is removed, and the polymer and nanoparticles are in solid state. Subsequently, the solids will heat up until the dissolution of the nanoparticle into the polymer starts.Since typical spray dryers in industry are not well isolated systems, the process can be assumed as non-adiabatic, meaning that the temperature will drop proportionally to the distance to the heater [26]. This heat loss is due to convection with the surroundings and heat radiation. The former one can be expressed as a function of the temperature gradient between inside the drying system and outside the system using Equation (11). The heat loss Q_loss_ is proportional to the contact surface between dryer and surrounding A, the specific thermal conductivity κ and the temperature gradient ΔT between the dryer and the surroundings [27]. Since the temperature gradient itself changes with the distance of the heater, its course is hard to predict:
(11)$$Q_{\mathrm{loss}\;}=\;\kappa A\Delta T$$


3.The energy balance of the spray dryer can be simplified as seen in Equation (12). The energy that is entering the spray dryer with incoming drying gas, E_in,_ is reduced to the outgoing energy, E_out_, by the energy needed for evaporation, E_vap._, for DisP, E_DisP._, and the heat loss due to convection with the surroundings, Q_loss_. All three are summarized to E_loss_ (Equation (13)):

(12)
$$E_{\mathrm{in}}={\;E}_{\mathrm{out}}+E_{\mathrm{loss}}$$


(13)
$$E_{\mathrm{loss}}=E_{\mathrm{vap}.}+Q_{\mathrm{loss}}+E_{\mathrm{DisP}}$$




4.The maximum temperature that the dried product can reach (T_max, product_) in this process, is the temperature of the drying gas after evaporation is completed, but before the DisP starts. This temperature becomes accessible by calculating an adiabatic process with $Q_{\mathrm{loss}}=0$. The resulting heat loss is presented in Equation (14). It is only dependent on the energy needed for the evaporation of water. Accordingly, T_product,max_ is the outlet temperature in case of an adiabatic process, without any heat loss due to DisP: $T_{\mathrm{out},\;\mathrm{adiabatic},\;\mathrm{before}\;\mathrm{DIS}.}$ (Equation (15)):


(14)
$$E_{\mathrm{loss},\mathrm{before}\;\mathrm{DisP}}={\;E}_{\mathrm{vap}.}$$


(15)
$$T_{\mathrm{product},\;\max}=T_{\mathrm{out},\;\mathrm{adiabatic},\;\mathrm{before}\;\mathrm{DisP}}\;\propto {\;E}_{\mathrm{in}}-E_{\mathrm{vap}.}$$




5.Calculating this adiabatic outlet-temperature can be done graphically with the Mollier-diagram or calculated using the underlying Equation (16). The later one was done in this work. C_p_ are the specific isobar heat capacities, $\dot{m}$ are the mass flows and $\Delta H_{\mathrm{vap}.H20}$ is the molar evaporation enthalpy of water [28]. The used parameters are listed in the Table 3.

(16)
$$T_{\mathrm{out},\mathrm{adiabatic}}=\frac{\left(c_{p}{\dot{m}}_{\mathrm{air},\mathrm{in}}T_{\mathrm{in}}\right)-\left(\Delta H_{\mathrm{vap},H20}{\dot{m}}_{H20,\mathrm{sample}}\right)}{\left(c_{p,\mathrm{air}\;}{\dot{m}}_{\mathrm{air},\mathrm{in}}\right)}$$



#### 2.2.8. Characterization Techniques

After nanomilling, the particle size was controlled using laser diffraction (LD) and dynamical light scattering (DLS) measurements. For LD measurements a Mastersizer 3000 by Malvern Panalytical, Worcestershire United Kingdom, equipped with a HydroMV cell was used (the unit was filled with demineralized water). Depending on the solid concentration of the suspension several drops were filled in the unit to achieve a laser shadowing of 6–10%. All samples were stirred during measurement at 1750 rpm and measured at room temperature. Three measurements were conducted using the Mie-theorem and the refractive index and density of the sample for evaluation: for indomethacin the refractive index was set to 1.546, the density to 1.2 g/cm³ [30]; for niclosamide the refractive index was set to 1.7, the density to 1.6 g/cm³ [31]; the prediluted sample out of the Mastersizer was used for DLS-measurements: ~0.5 ml was directly taken out of the HydroMV unit. A Zetasizer Nano ZS by Malvern Panalytical, Worcestershire United Kingdom, was used with disposable polystyrene cuvettes. The scattering was measured at an angle of 173° and automatic measurements (measurement position and attenuator were automatically chosen) were performed. Three measurements were done per sample. The same material parameters used for LD measurements were also used for the DLS analysis. The dried samples were then analyzed regarding their crystallinity using x-ray diffraction. Before the measurements, samples were placed in an agate mortar and ground to small particles. For obtaining X-ray diffractograms (XRD) a D2 phaser by Bruker, Massachusetts U.S, was used. The powder was placed on a lowered sample holder of a monocrystal of silicon and rotated during measurement at 10 rpm. X-ray beams were generated using a copper anode with 10 mA and 30 kV (the resulting wavelength was 1.54 Å). The scanning time was set to 0.5 s per datapoint and the angles 6–36° were investigated. The divergence slit was 0.2 mm, and the air-scatter knife was placed 1 mm above the sample.

## 3. Results

First experiments with indomethacin and PVP were performed to analyze the feasibility of NDM. Those experiments include the drying of a nanosuspension produced via milling and a microsuspension on a Kofler-bench to determine the necessary product temperature for amorphization. Thus, an aqueous nanosuspension was spray dried and compared to an ASD prepared via SD from organic solution. Furthermore, a complete “proof of concept” study was conducted to create an ASD out of crystalline niclosamide and PVP K25 as polymer.

### 3.1. Preliminary Experiments with Indomethacin

An indomethacin-PVP K12 suspension was comminuted until nanoparticles with D_x_10 = 75 nm, D_x_50 = 163 nm and D_x_90 = 432 nm were obtained. The particle size distribution (PSD) of the micro- and the nanosuspension is shown in Figure 4.

An aliquot of both, the microsuspension and the nanosuspension, was placed separately onto a Kofler-bench. In case of the nanosuspension the color changes to yellow after exceeding 110 °C; in case of the microsuspension this happens after reaching 140 °C. Samples of both experiments were taken at 120 °C and analyzed via XRD. The Kofler bench, the sampling positions and the resulting diffractograms are shown in Figure 5.

For SD of an indomethacin nanosuspension, PVP K25 was used instead of PVP K12. Otherwise, the composition remained the same. The resulting XRD is shown in Figure 6 with the freeze dried nanosuspension and a PM of the untreated raw material as negative control. The freeze dried nanosuspension still shows the characteristic peaks of indomethacin on e.g., 11.5°, 16.5° and at approx. 20° and 21°. The sample is obviously still crystalline, but the peak size is reduced. This could be due to the reduced sharpness of peaks in XRD that comes with small nanoparticles [32]. The spray-dried nanosuspension (C) shows no peaks, but just an amorphous halo. The sample seems to be XRD amorphous after the NDM process. The inlet temperature (T_in_) of 200 °C results in a maximum product temperature of 184 °C (with T_in_ = 200 °C, V_air,in_ = 18 m³/h, *p* = 1 bar, s = 6.06 g/kg air, ${\dot{m}}_{\mathrm{suspension}}$ = 1.5 g/min, x = 10%).

### 3.2. Proof of Concept

The PSD from the suspension before and after nanomilling measured by laser diffraction is shown in Figure 7.

After 45 min of comminution, the resulting suspension’s distribution parameters were: 21 nm (D_x_10), 63 nm (D_x_50), 190 nm (D_x_90). The absence of large particles with a diameter > 1 µm could be shown. The PSD was confirmed using DLS measurement, which resulted in D_x_10 = 58 nm, D_x_50 = 113 nm and D_x_90 = 310 nm.

The PdI was 0.24, indicating an acceptable narrow PSD. This means that nanomilling of the niclosamide suspension was successful.

For the proof of concept the following samples were prepared, and the XRDs of the produced compositions are presented in Figure 8:As negative control, a physical mixture of the components (“PM”, D in Figure 8);As positive control, an ASD prepared by solvent evaporation from organic solution (”ASD-SB”, A in Figure 8);Microcompositions which were spray dried with T_in_ = 220 °C and T_in_ = 120 °C (“Micro-DM”, G and H in Figure 8);Nanocompositions which were spray dried with T_in_ = 220 °C and T_in_ = 120 °C (“NDM”, B and C in Figure 8).

The diffractogram of ASD-SB (A) shows an amorphous halo indicating a pure amorphous substance and it confirms ASD-SB as positive control. The curve of PM (D) shows clear crystalline peaks which are assignable to the reference niclosamide anhydrate (F, grey). The curve of the nanosuspension spray dried at T_in_ = 120 °C (C) shows crystalline peaks assignable to niclosamide monohydrate (F, black) at 2Theta = 10° and 12°; peaks at 26° and 27° can be assigned to both niclosamide structures (F). The pattern of the peaks at 26° and 27° looks different to the one in the PM, which suggest a mixture of anhydrate and monohydrate. The curve of the nanosuspension spray dried at T_in_ = 220 °C (B) shows no residual crystalline peaks and a course comparable to the one of ASD-SB (A). This sample is XRD-amorph. Both curves of the microsuspensions spray dried with T_in_ = 120 °C and 220 °C (G, H) show crystalline peaks assignable to niclosamide anhydrate at 13°, 14° and at 26° and 27°. T_in_ = 220 °C corresponds to T_product,max_ = 209.7 °C which is 20 K below the melting temperature of niclosamide anhydrate and monohydrate (with T_in_= 220 °C, V_air,in_ = 25 m³/h, *p* = 1 bar, s = 6.06 g/kg air, ${\dot{m}}_{\mathrm{suspension}}$ = 1.5 g/min, x = 10%) [23]. It was possible to create an XRD-amorphous sample using NDM process and T_product,max_ = 209.7 °C. The corresponding measured outlet temperature was T_out_ = 90 °C.

### 3.3. Minimum Process Temperature

New niclosamide nanosuspensions were prepared and dried in a spray dryer. Thereby, T_in_ was varied to identify the lowest possible temperature at which an XRD-amorphous product can be produced. Afterwards the corresponding product temperature was calculated. T_in_ was varied between 170 and 220 °C. The resulting diffractograms are shown in Figure 9:

All diffractograms show a major amorphous halo. The sample dried at T_in_ = 170 °C shows multiple crystalline peaks at 2Theta = 12° and between 25° and 28°. At T_in_ = 180 °C and 185 °C, residual peaks are visible especially between 25° and 28°, which can be assigned to niclosamide monohydrate (F, black). At T_in_ = 190 °C only a small peak at 26.7° is left, but the peak at 11.8 ° shrunk. At T_in_ = 195 °C and above, the peaks all vanished. The samples were all XRD-amorphous. The lowest product temperature was calculated with T_in_ = 195 °C, ${\dot{V}}_{\mathrm{air},\mathrm{in}}$= 25 m³/h, *p* = 1 bar, s = 6.06 g/kg air, ${\dot{m}}_{\mathrm{suspension}}$= 1.5 g/min, x = 10%. The resulting T_product_ was 184 °C, which is 46 K below the melting temperature of crystalline niclosamide anhydrate and monohydrate [23]. The actual outlet temperature T_out_ measured by the spray dryer during the process was 85 °C. Furthermore, the residence time within the spray dryer t_r_ was calculated with following parameters: height = 47 cm, diameter = 15.9 cm, ${\dot{V}}_{\mathrm{air},\mathrm{in}}=$25 m³/h. The resulting volume of the drying chamber $V_{\mathrm{DC}}$ is 9.332 × 10^−3^ m³, and the resulting t_r_ is 1.34 sec. This is consistent with the specification for the Büchi spray dryer given by the manufacturer, who stated a t_r_ of 1–1.5 s [25].

### 3.4. Impact of Monohydrate Formation

NDM-experiments using a suspension containing the niclosamide monohydrate were performed to clarify the impact of the monohydrate formation. The synthesized monohydrate was codispersed with PVP K25 and SDS according to the composition shown in Table 2 and sprayed at T_in_ = 220 °C and 120 °C. The results are shown in Figure 10:

All curves still show crystalline peaks. The microsuspension containing niclosamide monohydrate did not undergo amorphization within the spray dryer, neither at T_in_ = 220 °C nor at 120 °C. According to Van Tonder, the dehydration of niclosamide monohydrate occurs at 100–120 °C when performed in a DSC at different heating rates [23], but the sample dried at 120 °C (B) still shows the characteristic peaks for the niclosamide monohydrate. For T_in_ = 120 °C, the temperature was too low to ensure dehydration. This was the case for the nanosuspension as well (Figure 8, C). The nanosuspension that was not amorphizable at T_in_ = 170 °C, shows peaks characteristic for the anhydrate (Figure 9, 170 °C). The required T_in_ for dehydration therefore lies somewhere in between 120 °C and 170 °C.

## 4. Discussion

### 4.1. Feasibility of Nano-Dry-Melting

To evaluate the NDM process in regard of its feasibility to form ASDs, a model system containing indomethacin and PVP was examined. Preliminary experiments on the Kofler-bench showed the differences between drying of a nanosuspension and drying of a microsuspension: in case of the nanosuspension, the dried material was amorphous after exceeding 120 °C, as proven by the corresponding XRDs. At said temperatures, the microsuspension was not able to form an ASD. In both cases, the temperature limit seems to be below the drug’s melting point (161 °C). The high solubility of indomethacin in PVP K12 probably resulted in a melting point depression, allowing the formation of an ASD [33]. In case of the dried nanosuspension, the temperature is in close approximation to the polymer’s glass transition temperature (110 °C). This indicates that the required temperature is here not (only) depending on the interaction between polymer and drug but on the viscosity of the pure polymer. After its T_g_ is reached, the viscosity greatly decreases, allowing the drug particles to dissolve into the polymer [34]. The pretreatment of the suspension in terms of nanomilling obviously reduced the subsequent necessary temperature for amorphization.

For SD of an indomethacin nanosuspension, PVP K25 was used instead of PVP K12, because of its higher pharmaceutical relevance. The higher T_g_ of PVP K25 (160 °C) implies good kinetic stabilization at room temperature, making it pharmaceutically more interesting than the lower viscosity grades (like K12), but more difficult to process using HME. NDM with the used settings was successful. However, since the real product temperature is probably somewhere between the calculated 184 °C and the measured outlet temperature of 80 °C, it cannot be said whether a melting or a dissolution process took place inside the spray dryer. If the temperature was above the melting point of indomethacin (160 °C), a melting process is more likely. Moreover, the untreated microsuspension was not dried with the same condition for comparison, and therefore it cannot be assumed that the complete NDM-process, including milling, was necessary.

Hence, a more suitable model system containing niclosamide anhydrate and PVP K25 with a high T_g_ of 160 °C was examined. Thermal investigations revealed its high melting point of 230 °C and its high tendency to recrystallize making it a suboptimal candidate for processes like HME and an interesting candidate for NDM. Milling of niclosamide in the WSMM resulted in a very narrow PSD with particle sizes with a D_x_50 < 100 nm. The outcome of LD indicated smaller particles than those measured via DLS. A possible explanation could be the influence of the shape and orientation of the particles. Van Tonder provided SEM pictures of the different crystal forms of niclosamide, showing that all three investigated forms have a needle-like shape [23]. The results show that the comminution was successful to a certain extent (nano-scaled at least in the needles’ diameter), but the PSD should be evaluated using, e.g., electron microscopy (like SEM) to be definite regarding the size and shape of the nanoparticles. DM in the spray dryer provided several conclusions, including the successful amorphization at 210 °C, which is below the melting temperature of niclosamide (230 °C). The necessity of the nanomilling was proven by processing the microsuspension without subsequent nanomilling. The same temperature exposition did not lead to an XRD amorphous sample.

The resulting distances in the niclosamide compositions were calculated and compared. The used densities of the components are: 1.6 g/cm³ for pure drug, 1.2 g/cm³ for pure polymer, and 1.26 g/cm³ as mix density. In both cases, a drug load of 20 wt.-% (which is typical for ASDs) and exemplary spray-dried particles consisting of the ASD (“ASD-particles”) with a diameter of 100 µm were employed. The measured D_x_50 values were used: 63 nm in case of the nanocomposition, 17.8 µm in case of the microcomposition. First, the volumetric drug load needs to be calculated with the volume of the hole ASD particle, $V_{\mathrm{ASD}}$, and the part of this particle that consists only of drug, $V_{\mathrm{drug}\;\mathrm{per}\;\mathrm{ASD}\;\mathrm{particle}}$ (Equation (17)). The latter one was calculated using the mass of one ASD particle, $m_{\mathrm{ASD}\;\mathrm{particle}}$, the drug load in wt.−%, ${\;\mathrm{DL}}_{\mathrm{wt}.-\%}$, and the density of the pure drug, $\rho_{\mathrm{drug}\;}$ (Equation (18)):(17)$${\mathrm{DL}}_{\mathrm{vol}.-\%}=\frac{V_{\mathrm{drug}\;\mathrm{per}\;\mathrm{ASD}\;\mathrm{particle}}}{V_{\mathrm{ASD}}}=15.8\;\%$$
(18)$${\;V}_{\mathrm{drug}\;\mathrm{per}\;\mathrm{ASD}\;\mathrm{particle}}=\left(m_{\mathrm{ASD}}{\mathrm{DL}}_{\mathrm{wt}.-\%}\right)\rho_{\mathrm{drug}}$$

Filling the ${\mathrm{DL}}_{\mathrm{wt}.-\%}$ and the radii into Equations (6) and (7), the distances can be calculated. The calculated IPS are 62 nm in case of a nanocomposition and 17,488 nm in case of a microcomposition. The maximum distances of their centers are 188 nm for the nanoparticles and 53,088 nm for the microparticles. Both distances are reduced by approximately a factor of 300 which corresponds to the reduction in d. The results are visualized in Figure 11. As a result, the time required for the homogenization of the already dissolved molecules, according to Equation (9), is decreased quadratically related to the particle size. This difference in spacing alone has a high impact on the required processing time, even though the additional benefit of the enhanced dissolution rate according to the Noyes-Whitney equation is not even considered at this point. In conclusion, it can be said that the time required to produce a homogeneous ASD is at least drastically reduced due to the accelerated time for homogenization. The theoretical background promises further reduction due to the favorable dissolution properties of nanoparticles., accelerating the process time of NDM greatly.

Unexpectedly, the peaks in the diffractogram of the dried nanosuspension compared to those of the dried microsuspension indicate different crystalline (pseudo-)polymorphs. The nanosuspension showed peaks assignable to the monohydrate, whereas the microsuspension showed peaks assignable to the anhydrate only. The nanosuspension obviously underwent hydration throughout or after the nanomilling process. The influence of the monohydrate formation was investigated by processing of the monohydrate in its microscopic scale. The DM was not successful with monohydrate particles at sizes in the micrometer range. This raises the question of whether, in case of NDM, the nanoparticles were dehydrated prior to the onset of DisP, or whether DisP occurred instead of the dehydration process. If the former one was the case, it could be speculated that the dehydration works as an initiator for DisP. During removal of water from the crystal unit, the lattice structure is temporarily disrupted which again means that the energetic state of the lattice is impaired. Thus, a lower lattice energy must be overcome to start the dissolution process. This would lead to a more spontaneous, accelerated amorphization process [35]. Using different analytical techniques that can differentiate whether the residual crystallinity at T_in_ = 180 °C originates from the anhydrate or the monohydrate could provide further hints regarding this hypothesis. Furthermore, experiments at which the nano-scaled anhydrate of niclosamide is used could be conducted to explore the necessity of the monohydrate formation during NDM process.

Nevertheless, the hydration was definitely not the sole reason for the successful dissolution of the nanoparticles. It is evident that for NDM, the presence of nano-scaled drug particles is crucial. The reason for the successful DM can be attributed to the energy introduced during milling that led to the size reduction. In the case of niclosamide, this energy probably also led to side benefits like extensive intertwining between polymer and drug particle, surface effects on the particles [36], and the hydrate formation of the drug [35].

The successful amorphization of both model systems led to further research regarding the necessary thermal stress exposed to the drug system.

### 4.2. Nano-Dry-Melting as Novel, Alternative Process Technology

To evaluate the process advantages over state-of-the-art techniques, the thermal stress to the substance was determined. Thereby, t_r_ and T_product,max_ were evaluated.

In a common SD process the process temperature is sufficiently high to evaporate the solvent, but as low as possible to minimize the thermal load on the drug product. In most cases the outlet temperature of the drying gas is therefore used as an estimation for the maximum product temperature [37]. In the case of the NDM process, the spray dryer is not only removing the solvent but should also enable DisP. This leads to different necessary process settings that allow a very fast evaporation of water and enable the dry product to experience thermal load inside the system. Table 4 compares the process setting of a conventional SD process to that one used for DM:

In the case of NDM using a spray dryer, the outlet temperature of the drying gas is no suitable parameter for the product temperature, as the product is already completely dry while still in the system; thus, a different estimation was made. During DM, the product undergoes different modifications. The nanosuspension will undergo the transition to a nanocomposition (due to the evaporation of the liquid), and furthermore to a completely amorphous system.

The course of the temperature within the spray dryer (Figure 12) was assumed to be as followed: During the first step (1), the temperature drop is high due to evaporation of water. When “T_out,adiabatic,before_
_DisP_” is reached, all water is evaporated, but the solids are still at T = 100 °C. Thereafter, the drying gas will heat up the nanocomposition (2) until it approaches its own temperature (indicated by the black circle). Depending on the specific heat capacity of the polymer this needs a certain time period. Thereafter, the DisP of the NDM process of the product can start (3). The approach used here calculates T_out,adiabatic,before DisP_ (which is between step 1 and 2). With further knowledge about the specific heat capacity of the product, an even more exact T_product,max_ could be estimated (indicated as circled dot). It must be kept in mind that in contrast to this adiabatic model, extensive heat loss occurs in spray drying systems. Therefore, the calculated temperature is only an estimation and the real T_product,max_ is certainly lower which means less thermal load on the product. Accordingly, the real T_product,max_ lies somewhere in between this calculated T_out,adiabatic,before DisP_ and the measured T_out_.

The lowest calculated T_product,max_ sufficient for complete amorphization of the niclosamide- PVP K25 system was 184 °C, which was 46 K below the melting point of niclosamide (230 °C).

Since the feed rate during SD was very low (1.5 g/min), it can be assumed that any heat loss due to the NDM and evaporation process was relatively small. The huge difference between the calculated T_out,adiabatic,before DisP_ of 184 °C and the real T_out_ of 85 °C therefore suggests a high nonadiabatic heat loss to the surroundings. According to Equation (11), this heat loss is at its maximum directly after the inlet, where the difference between temperature in the system and temperature outside is the highest. Consequently, a major part of the nonadiabatic heat loss probably occurred before DisP could begin. The remaining temperature for vaporization and DisP according to Equation (15) was most likely lower, which would result in a lower T_product,max_.

The time of exposure was no more than 1.34 s. However, this estimation is only applicable under the assumption of a complete laminar flow in the spray dryer. In practice, turbulences within the apparatus will most likely prolong this time. Moreover, this estimation will lead to the residence time of the aqueous suspension. The exposure time of the dried product is reduced by the time period necessary for the evaporation of the water. It was assumed that the prolonging impact of the turbulences and the shortening effect of the water evaporation will equalize each other. Therefore, this first approach using the residence time of the drying gas should still enable the estimation of a meaningful value for the heat exposure time of the dried product. There is literature available, however, where the residence time in the spray dryer used here under similar conditions was captured with a tracer experiment. The measured median residence time of the product was 6 sec in the work by Schmitz-Schug [39]. Tracer experiments using our system and settings could clarify the exact heat exposure time for niclosamide and PVP [40].

Compared to HME, where t_r_ of 5 to 10 min or even more are standard [41], this method provides amorphization with significant lower exposition to thermal stress. Additionally, no mechanical forces were added during the drying processes. Future work could directly compare the heat stress exposed to ASDs of the same substance system produced via HME and NDM. Experiments where degradation products of the polymer/drug substance are carefully monitored could be considered to give exact numbers for comparison of both methods.

As a result, NDM can open a design space that was not accessible using HME by lowering the required temperature to start the dissolution and enabling a faster homogenization to complete the process. Such requirements can be used to display the design space in the system’s phase diagram [42]. In Figure 13, two cases of phase diagrams are illustrated, which represent the thermodynamic (upper pictures) and kinetic situation (lower pictures) for a thermostable (case 1) and thermosensitive system (case 2). HME is only applicable at temperatures at, or just slightly below, the drug’s melting temperature. In case of a thermosensitive system, this applicability of HME is strongly restricted by the degradation kinetics of the drug/polymer. Since for HME a certain heat exposition time is required, the design space (green area) is limited by the area in which degradation occurs (red), and by the area in which the kinetic of the compound is limited (black). If the degradation already begins within this design space, processing with HME becomes difficult [42]. In case of NDM, this residence time is highly reduced as explained above. Even systems with thermosensitive components are accessible, therefore the available design space enlarges to lower process temperatures. The strong dependence of HME on the melting point and the degradation temperature of drug and polymer is therefore at least partially overcome and further design space is opened, depicted by dark green region.

In contrast to state-of the art amorphization approaches, NDM represents a method with two clear advantages for industrial use:It only requires water and no organic solvents (in contrast to solvent-based methods), which is advantageous for economic as well as for ecological reasons;It exposes the sample to lower temperatures for a shorter time (in contrast to fusion-based methods) which minimizes thermal stress to the product.

One has to consider that this process technology requires two unit operations, the formation of the nanosuspension and the subsequent Dry-Melting step. Hence, expertise in various technologies would be advantageous or a brief process development step. However, NDM consists of already established and approved unit operations like milling and drying. Hence, typical process operations can be directly inherited. The most important requirement is probably the possibility of the system to be converted into a nanosuspension. Fortunately, the NDM process is very adaptable regarding its unit operations. The exact mechanism of forming this nanosuspension can be varied according to the properties of the system. For example, a very unstable nanosuspension would be preferably created via a top-down rather than a bottom-up mechanism. The Dry-Melting step can be performed using various process techniques as well, allowing the adjustment of residence time, required temperature or in general the drying mechanism. The most promising ones include SD as a fast and simple option, fluidized bed drying and contact drying, using, e.g., a drum dryer as an alternative in case of insufficient residence time; also, extruder-based processes are feasible for drying [43]. Furthermore, NDM has potential to be conducted in continuous manufacturing, following the actual pharmaceutical trend [44].

### 4.3. Analytical Considerations

Additional methods to ensure the amorphicity like fourier-transform infrared spectroscopy (FTIR) and DSC were considered. Since FTIR is based on qualitative comparison of spectra, the raw material, a positive control ASD and a nanocomposition should be available [45]. This was not possible in the case of niclosamide, since complete amorphization of the pure substance was not achievable. DSC thermograms of the monohydrate show a dehydration peak at 120 °C [23]. This temperature was suspected to be in the same range as the solubility temperature. The use of the melting endotherm to identify residual crystallinity was therefore disregarded. Other additional analytical methods like atomic force microscopy should be studied to ensure the complete amorphicity [46]. Another issue is the homogeneous distribution of the amorphous material. Inhomogeneities could lead to a decreased stability since drug-rich regions are locally not stabilized by the necessary amount of polymer, and could act as nuclei which cause recrystallization of the drug [47]. However, since for nanoparticles the diffusion path length that is necessary for complete homogenization is >100 times smaller compared to the case when microparticles are being used, a complete homogenization can be assumed. Furthermore, there is an indirect hint that homogenization is reached during NDM processing: it was shown that even such a strong and fast crystallizer like niclosamide could be successfully processed using a spray dryer. One can hypothesize that if there would be drug-rich regions, recrystallization should happen quickly in case of a class-I crystallizers, which was not observed within four weeks of storage.

## 5. Conclusions

NDM was proven to successfully create ASDs by exploiting the unique properties of nanoparticles. Due to the high surface area and thinner diffusion layer, nanoparticles can dissolve very rapidly in a polymer matrix Furthermore, the distance to travel to reach homogenization is, for well distributed nanoparticles, strongly reduced (>100 times) compared to the case when microparticles are used. Both effects lead to a very short processing time even at reduced temperature. This was demonstrated in indicating experiments with indomethacin PVP K12, and further investigated with niclosamide-PVP K25. Temperatures below the melting point of the crystalline drug and short exposure times of a few seconds were sufficient to create XRD-amorphous samples. This was only possible by comminution of the particles in a WSMM to a size in the order of 100 nm, as comparisons to the Dry-Melting process of untreated microsuspensions showed. It is supposed that independent of the drug model systems, niclosamide and indomethacin used in this study the NDM process are generally applicable if the drug is comminuted in the submicron regime.

NDM provides several benefits while causing minimal disadvantages. This makes the NDM process attractive to produce ASD with those drugs that either are insufficiently soluble in pharmaceutically acceptable ICH class 3 solvents or have a high melting point, which includes the risk of thermal degradation during processes like HME.

## Figures and Tables

**Figure 1 pharmaceutics-14-02145-f001:**
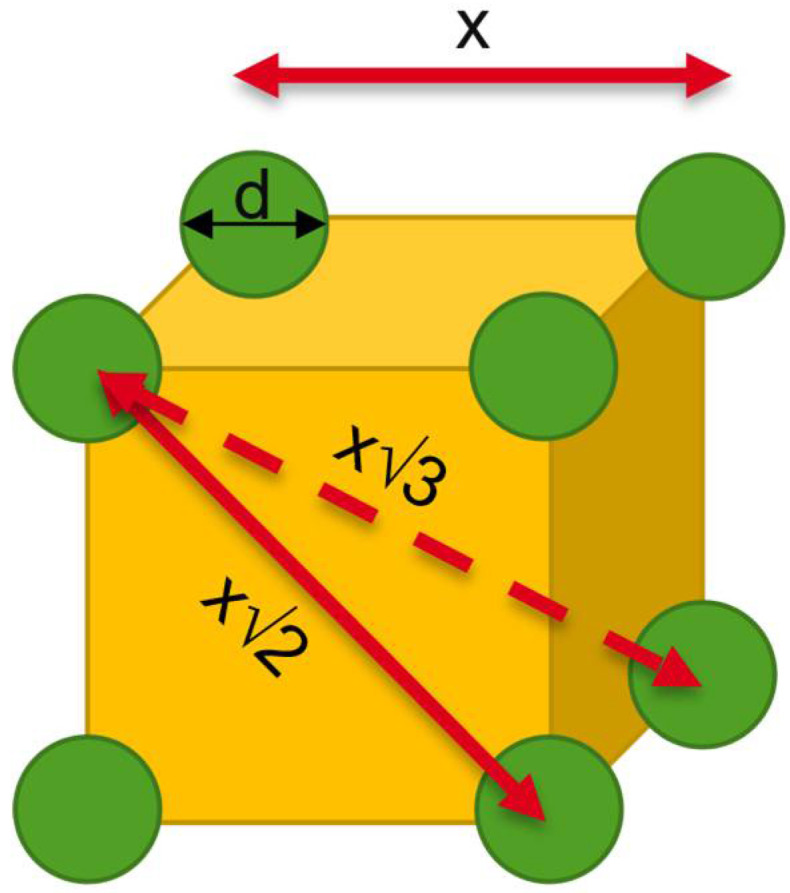
Model of a cubic grid to illustrate the distribution of particles inside a Nano-/Microcomposition. Green spheres = drug particles with diameter d, yellow = polymer matrix.

**Figure 2 pharmaceutics-14-02145-f002:**
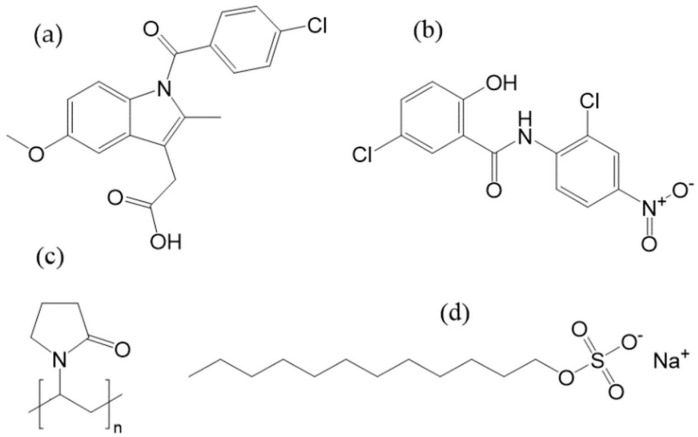
Chemical structure of (**a**) indomethacin, (**b**) niclosamide, (**c**) PVP and (**d**) SDS.

**Figure 3 pharmaceutics-14-02145-f003:**
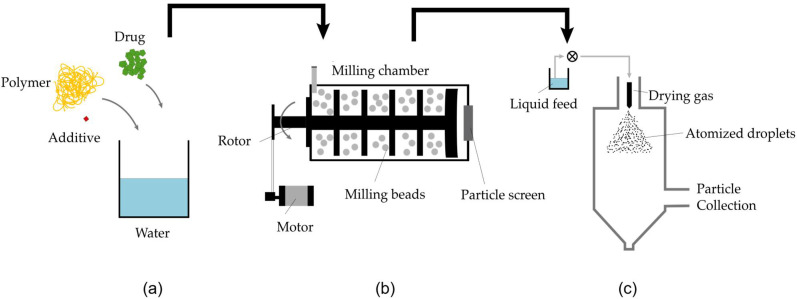
Nano-Dry-Melting process consisting of (**a**) Preparation of suspension. (**b**) Comminution in a wet stirred media mill and (**c**) Dry-Melting (DM) in a spray dryer, which includes the solvent evaporation and the dissolution of drug into polymer.

**Figure 4 pharmaceutics-14-02145-f004:**
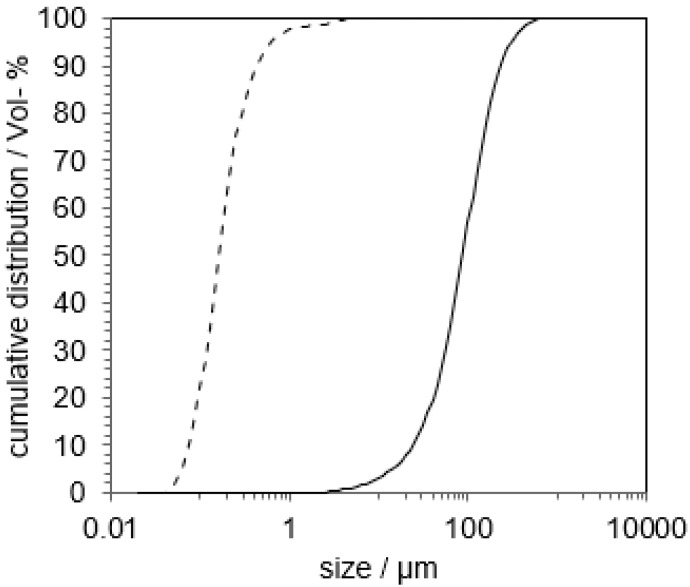
Cumulative particle size distribution of indomethacin suspension before (─) and after milling (- - -). Measured via laser diffraction.

**Figure 5 pharmaceutics-14-02145-f005:**
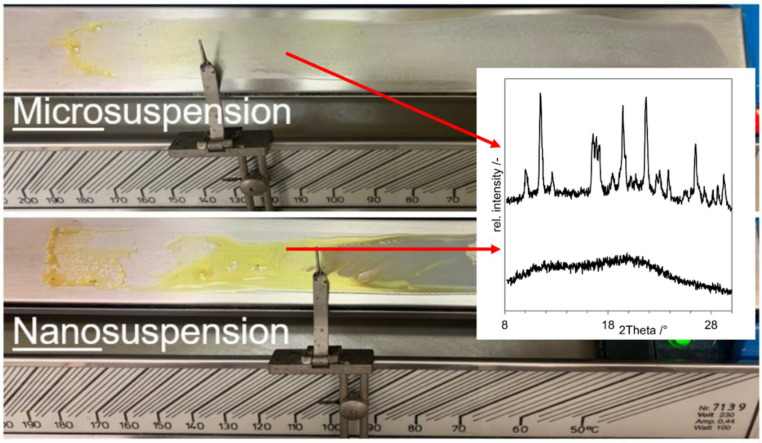
Kofler-bench experiments with indomethacin suspensions. The upper picture displays the result of the drying of the microsuspension. The lower picture displays the drying of the nanosuspension. The arrows indicate at which positions samples were taken. Corresponding X-ray diffractograms are shown on the right.

**Figure 6 pharmaceutics-14-02145-f006:**
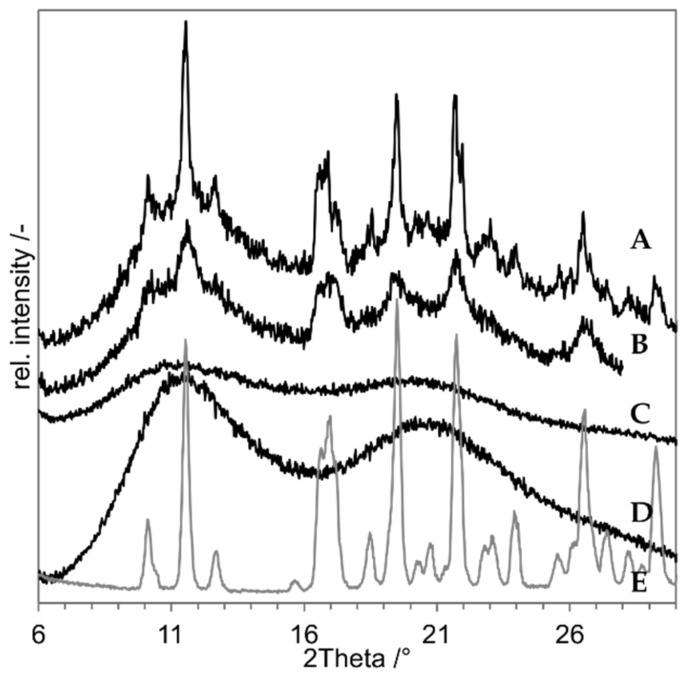
X-ray diffraction results of spray drying of indomethacin nanosuspension. (A) Physical mixture (negative control), (B) Freeze-dried Nanosuspension (negative control), (C) Nano-Dry-Melting at T_in_ = 200 °C, (D) PVP K25, (E) Indomethacin.

**Figure 7 pharmaceutics-14-02145-f007:**
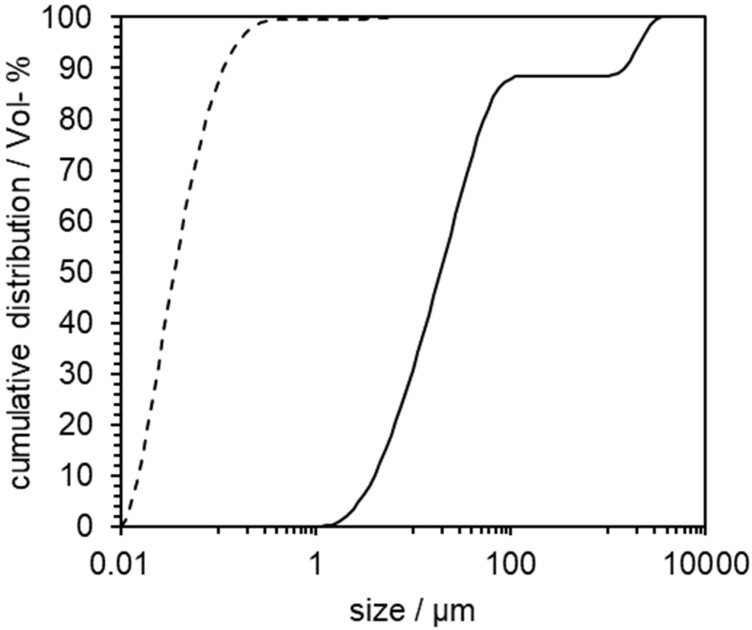
Cumulative particle size distribution of niclosamide suspension before (─) and after milling (- - -). Measured via laser diffraction.

**Figure 8 pharmaceutics-14-02145-f008:**
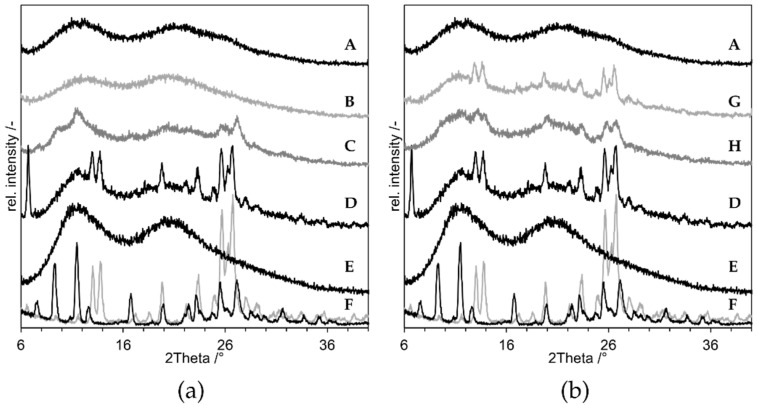
X-ray diffraction results of “proof of concept” experiments with niclosamide. (**a**) the results of the nanocompositions, (**b**) results of the microsuspension are displayed. A: ASD-SB (positive control), B: NDM at 220 °C, C: NDM at 120 °C, D: PM (negative control), E: PVP K25, F: niclosamide anhydrate (grey), monohydrate (black), G: Micro-DM at 220 °C, H: Micro-DM at 120 °C.

**Figure 9 pharmaceutics-14-02145-f009:**
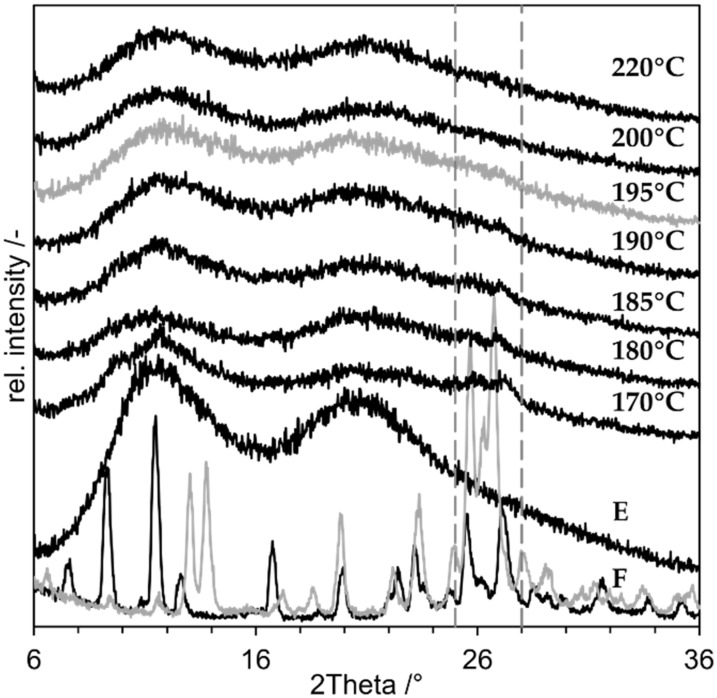
XRD-results of niclosamide nanocomposites produced by NDM with varying T_in_ of the spray dryer. T_in_ is depicted on the right, E: PVP K25, F: niclosamide anhydrate (grey), monohydrate (black). The dotted grey lines mark the angels 25–28°.

**Figure 10 pharmaceutics-14-02145-f010:**
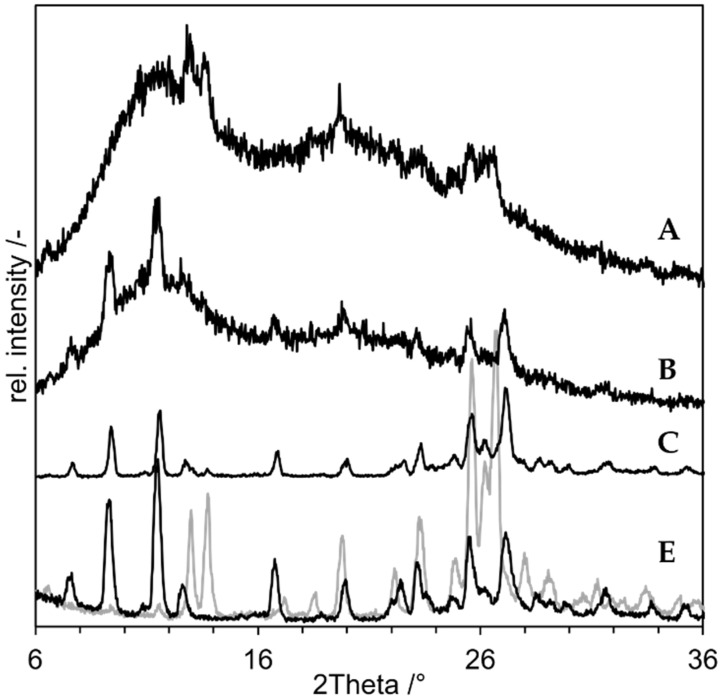
X-ray diffractograms of Dry-Melting experiments with microsuspension containing niclosamide monohydrate. (A) Suspension dried at 220 °C, (B) Suspension dried at 120 °C, (C) Suspension dried at room temperature, (E) Niclosamide anhydrate (grey), Niclosamide monohydrate (black).

**Figure 11 pharmaceutics-14-02145-f011:**
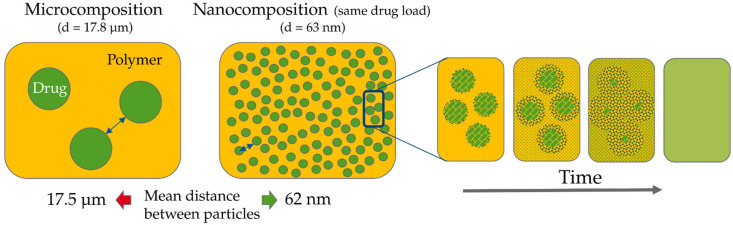
Distances of particles in the niclosamide micro- vs. nanocomposition with 20% DL on the left. The polymeric matrix is depicted in yellow. The green spheres correspond to the drug particles. The displayed distances correspond to the path length between the surfaces of the particles. On the right the dissolution and homogenization process are illustrated schematically.

**Figure 12 pharmaceutics-14-02145-f012:**
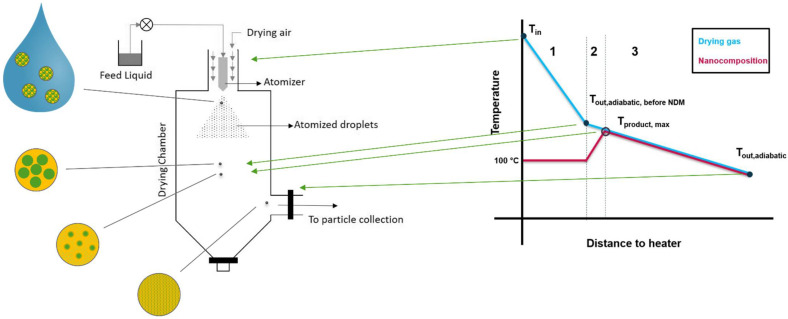
Nano-Dry-Melting (NDM) Scheme inside a spray dryer (adapted from [38], MDPI, 2019) connected to the simplistic temperature course during NDM. Blue: course of the drying gas temperature withing the spray dryer. Red: Course of the temperature of the nanocomposition. The courses are divided into three parts: 1: Evaporation of water. 2: Heating up of the Nanocomposition. 3: the Dry-Melting process. Depicted in yellow is the polymer matrix, the drug particles are in green.

**Figure 13 pharmaceutics-14-02145-f013:**
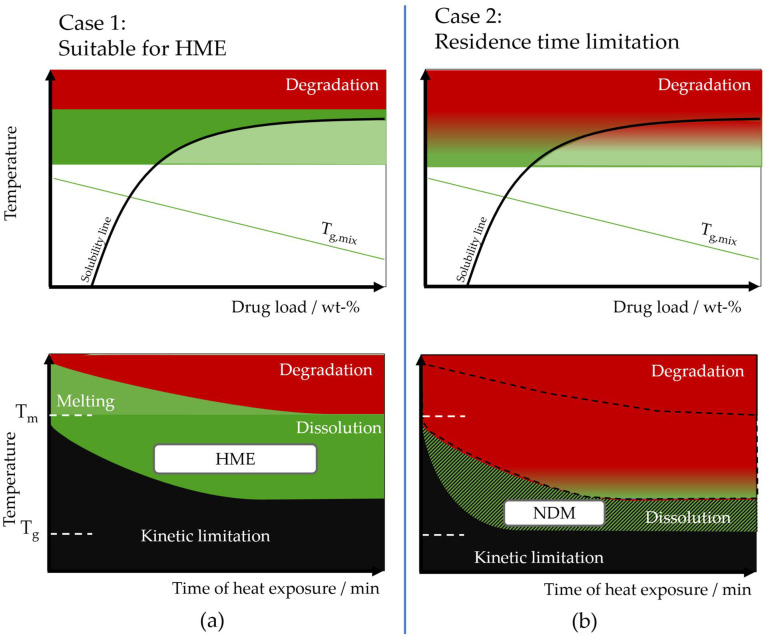
Principal schematic of the different thermodynamic (upper pictures) and kinetic (lower pictures) situation during ASD processing. (**a**) Case 1 with a suitable design space that allows Hot Melt Extrusion (HME) processing. (**b**) Case 2 with the phase diagram of a thermosensitive system. Early degradation kinetically limits HME processing. The available design space in case 1, marked by the black dashed line is red in case 2 because of degradation. Nano-Dry-Melting (NDM) is capable of opening further design space due to the short heat exposure times required. Depicted in green is the available design space, the green, black hatched area is showing the additional design space available through NDM. In the red area degradation occurs, the black area is unavailable for processing due to kinetic limitations.

**Table 2 pharmaceutics-14-02145-t002:** Composition of Indomethacin and Niclosamide Suspension.

Ingredient	Function	Indomethacin Suspension		Niclosamide Suspension	
		Concentration in Suspension/wt.-%	Concentration in Solid/wt.-%	Concentration in Suspension/wt.-%	Concentration in Solid/wt.-%
Indomethacin	Drug	6.6	33		
Niclosamide	Drug			1.89	19.7
PVP K25	Stabilizer			7.56	78.8
PVP K12	Stabilizer	13.2	66		
SDS	Surfactant	0.02	1	0.15	1.5
Water	Fluid	80	-	90.40	-

**Table 3 pharmaceutics-14-02145-t003:** Parameters for calculating the outlet temperature.

Term	Description	Value	Reference
** $c_{p,air}$ **	Specific heat capacity of air	$1005\;\frac{J}{\mathrm{kgK}}$	[29]
** $\Delta H_{vap,H20}$ **	specific enthalpy of vaporization of water	225,700 $\frac{J}{\mathrm{kg}}$	[29]
** ${\dot{m}}_{air,in}$ **	Mass flow of drying gas	Depending on experimental setup	${\dot{m}}_{\mathrm{air},\mathrm{in}}={\dot{V}}_{\mathrm{air},\mathrm{in}}{\;\rho }_{\mathrm{air},\mathrm{in}}$ $\rho_{\mathrm{air},\mathrm{in}}=\frac{{\mathrm{pM}}_{\mathrm{air},\mathrm{wet}}}{{\mathrm{RT}}_{\mathrm{in}}}$ $M_{\mathrm{air},\mathrm{wet}}={\;M}_{\mathrm{air},\;\mathrm{dry}}+s$
** ${\dot{m}}_{H20,\;sample}$ **	Mass flow of water in feed	Depending on experimental setup	${\dot{m}}_{\mathrm{suspension}}\left(1-x_{\mathrm{solid}}\right)$
** $T_{in}$ **	Inlet temperature of drying gas	Depending on experimental setup	Direct setpoint on spray dryer

Wherein M = molecular mass in g/mol, $\dot{V}$ = Volume flow in m³/ h, $\dot{m}$ = Mass flow, ρ = density, R = ideal gas constant (8.314 J/molK), s = specific humidity in g water /kg air, x = solid fraction.

**Table 4 pharmaceutics-14-02145-t004:** Conventional SD vs. SD for Dry-Melting.

Property	Conventional SD	SD for DM
Temperature	As low as possible T ~ T_vap.H20_ *	As high as necessary T ~ T_g_ > T_vap.H20_ *
Residence time	Shortest residence time possible	Sufficiently long residence time for Evaporation + DisP
Droplet formation	Gentle nozzle pressure to generate big droplets resulting in big particles with good flowability properties	Very high nozzle pressure to create small droplets and enable fast evaporation
Residual moisture	Generally higher due to slow evaporation rate, Residual moisture can be removed in additional drying process	Generally low since evaporation rate should be high

* T_g_ = glass transition temperature, T_vap.H20_ = evaporation temperature of water.

## Data Availability

Not applicable.

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
