# Peer review of "Nano-Dry-Melting: A Novel Technology for Manufacturing of Pharmaceutical Amorphous Solid Dispersions"

_pharmaceutics, 2022, doi:10.3390/pharmaceutics14102145_

Round 1

Reviewer 1 Report

The manuscript pharmaceutics-1920312 “Nano-Dry-Melting: A Novel Technology for Manufacturing of Pharmaceutical Amorphous Solid Dispersions” was reviewed. This paper reported a new manufacturing technology “Nano-Dry-Melting” with combination of nano-crystal formulation and spray drying for amorphous solid dispersion formulation. The drug with nano-crystalline state dissolved into a polymeric carrier with relative lower temperature during spray drying process, which enabled to formulate amorphous solid dispersion for a brick dust compound like niclosamide. This research is worthy for publication of Pharmaceutics “Special Issue: Amorphous Solid Dispersions: Rational Selection of a Manufacturing Process”. And, the manuscript is well written. The only minor comments were listed below for the minor revision.

1. Table 1 summarizes the Pros/Cons of the preparation methods for amorphous solid dispersion. This is important and make readers understood for this technology. In this paper, the authors studied NDM as a new process. Hence, I recommend the authors to summarize advantage, disadvantage and requirements for processability of NDM in discussion or conclusion (e.g. Application of NDM is limited for the drugs which can form nano-crystal).

2. Different abbreviation of horus is used; for example, 3 h (line 217),  but 3 hours (line 218). Please use same abbreviation through the manuscript.

3. The procedures of suspension and/or nano-milling and spray-drying was different between the two drugs. Pleas add brief explanation of this reason for readers.

4. “Material” was written following “Technical implementation of NDM process”. However physicochemical properties of the drug and the excipients were given in “Material”. So, I believe that “Material” should be written prior to “Technical implementation of NDM process”.

5. PVP K12 (~110°C) is PVP K12  (Tg~110°C)? (line 330)

6. The reflective index and the density of niclosamide was given from the reference 34 (line 350), but the reference of those of indomethacin was not shown (line 349). Please add it.

7. The (PSD) in line 419 should be corrected.

8 The particle size of the drugs following the nano-milling was shown as the LD data. Where the authors showed the DLS data?

9. The authors carefully discuss the heat stress and time during the process and the advantage of NDM was shown. This is very worthy for the compound with lower solubility and higher melting point as brick dust molecule. The calculated value of temperature during NDM (spray drying of nano-crystal) was given there. I believe that direct comparison between NDM and HME using various drug and polymer from viewpoint of heat stress (e.g. amorphization and thermal degradation) in further study. I the authors agree it, please add those viewpoint in the discussion or the conclusion.

That’s all.

Reviewer 2 Report

The work describes a novel process technology, Nano-Dry-Melting (NDM), as a means to solve the limitations presented by the thermal and mechanical shear stresses or solvent use during industrial production/use of ASD. The technique has potential for industrial application.

The methods of ASD production are very well presented and summarized in the introduction. The following sections are less clear and at times English language and presentation should be improved.

In general, the manuscript reports relevant findings but it needs to be reorganized (see below); a list of the points, which need to be addressed follows:

L23 – exposition to what? Please clarify.

L40 – consider using ‘roughly’ instead of ‘superficially’.

L75-77 – please explain or reformulate the sentence, which doesn’t seem to make sense: ‘Therefore …. are formulated [3]’.

L79-80 – which ‘keywords’ are you referring to? are they presented in Table1? Please review the sentence.

Table 1 – please consider using a footnote to explain abbreviations, especially those which are not given in full in the text or appear for the first time (e.g. HSEQ, PSD). Spray drying, for example, has no disadvantages or processing requirements? Complete and do not leave blank cells in the table; if you have nothing to say, please use ‘–‘ instead; Table needs editing – continuation should appear on a different page.

L98 – Section 2 provides the background of the work and should, in fact, be part of the introduction. Though uncommon, you could present it as a subsection within the Introduction. Furthermore, this section is lengthy and could be shortened, with parts of the text moved into the Discussion (e.g. text in L195-203 is clearly Discussion) or even the Materials and Methods, when equations for calculations made are given. Whatever solution you come up with, Section 2 should be Materials and Methods.

L111 – Any reason why K is used in some temperatures? Please consider substituting by °C throughout the manuscript; ensure consistency.

L134, 135 – DIS? dissolution? Please make sure that every abbreviation is given in full at first mention.

L151 – the model does not ‘visualize’; it allows ‘visualization’.

L193 – IPS?

Materials and Methods – Please give model, maker, city, and country for every piece of equipment used (e. g. wet stirred media mill, freeze-dryer, mill – L226 – etc.); subsections should be numbered (2.1 Materials, 2.2 Methods; these should follow one another and be clearly separated). Comparison of methods should not be given in this section (e.g. Table 4 and text that follows) but discussed with the results or in the introduction. In this section the methods should be presented succinctly; if further explanation/discussion is needed, I suggest that you present it in Supplementary Material. Likewise, justification on the use of materials (e.g. details on the niclosamide use – L320-323) should be moved into the corresponding part of the discussion.

Tables 1 and 2 should be combined.

L268 and Table 4 – DM?

Table 5 – please consider substituting ‘origin’ with ‘reference’; check that all the abbreviations are in the footnote.

L419 – why is PSD in-between parenthesis?

Discussion is presented as an autonomous section; however, results have already been discussed while presented. Please consider combining Results and Discussion in a single section; this would avoid unnecessary repetitions. Subsections should be numbered.

L628-630 – please clarify sentence ‘But it is worth knowing that there is literature available, where the residence time in the here used spray dryer under similar conditions was measured with tracer experiments'.

L631 – the median residence time given was supposedly determined during the present work. Therefore, why is a reference [41] given?

Conclusions’ section should be brief reporting only the main conclusions from the work and not discuss the work again, as done. Discussion of results should go to the corresponding section.

Reviewer 3 Report

The manuscript written by Hermeling, et al reported the proof of concept and feasibility study of nano-dry-melting which is a novel approach to produce amorphous solid dispersion. This new approach can provide advantage than normal approach such as it doesn’t require organic solvent in the process and potential reduction in temperature which minimize thermal stress to the amorphous solid dispersion product. The manuscript is well written and I would like to recommend this paper for publication in Pharmaceutics. However, there are some points that authors should address to increase clarity and attract readership. Herewith, I listed my point-to-point comment

1.      One of the important aspect in developing ASD is its stability. As the particle size can be reduced significantly, it will increase the entropy of the system which may lead to instability of the amorphous system. Has the author explored stability issue of ASD, ie stability under accelerating aging condition 40C/75%RH? You can give the comment regarding this issue without further inclusion of new data.

2.      I don’t really understand the meaning of vertical line between (b) and (c) in Figure 2. To some extent it shows clear boundary between WSMM and DM in spray drying, but author mentioned that this can be also continuous process in another part of the manuscript. My guess is these pictures are taken from existing picture as the format is slightly different such as rather 3D in (b) and 2D in (c). I would recommend to redraw the picture in order to increase clarity. This paper contains new method to produce ASD, so it will be good if it is accompanied by better illustration as well.

3.      Is there any motivation to use different PVP in indomethacine and niclosamide?

4.    ”X” annotation should be use in multiplication instead of “*” annotation in all equations, ie equation 16

Round 2

Reviewer 2 Report

The suggestions made to the authors' have been followed and the manuscript now seems acceptable for publication.